# Optimizing Production, Characterization, and In Vitro Behavior of Silymarin–Eudragit Electrosprayed Fiber for Anti-Inflammatory Effects: A Chemical Study

**DOI:** 10.3390/bioengineering11090864

**Published:** 2024-08-25

**Authors:** Foram Madiyar, Liam Suskavcevic, Kaitlyn Daugherty, Alexis Weldon, Sahil Ghate, Takara O’Brien, Isabel Melendez, Karl Morgan, Sandra Boetcher, Lasya Namilae

**Affiliations:** 1Department of Physical Science, Embry Riddle Aeronautical University, Daytona Beach, FL 32114, USA; 2Department of Human Factors and Behavioral Neurobiology, Embry Riddle Aeronautical University, Daytona Beach, FL 32114, USA; 3Department of Electrical Engineering, Embry Riddle Aeronautical University, Daytona Beach, FL 32114, USA; 4Department of Mechanical and Engineering Sciences, Embry Riddle Aeronautical University, Daytona Beach, FL 32114, USA; 5Seminole High School, 2701 Ridgewood Ave, Sanford, FL 32773, USA

**Keywords:** drug–polymer complex, electrospray, inflammatory bowel disease, microencapsulation, silymarin

## Abstract

Inflammatory Bowel Disease (IBD) is a chronic condition that affects approximately 1.6 million Americans. While current polyphenols for treating IBD can be expensive and cause unwanted side effects, there is an opportunity regarding a new drug/polymer formulation using silymarin and an electrospray procedure. Silymarin is a naturally occurring polyphenolic flavonoid antioxidant that has shown promising results as a pharmacological agent due to its antioxidant and hepatoprotective characteristics. This study aims to produce a drug–polymer complex named the SILS100-Electrofiber complex, using an electrospray system. The vertical set-up of the electrospray system was optimized at a 1:10 of silymarin and Eudragit^®^ S100 polymer to enhance surface area and microfiber encapsulation. The SILS100-Electrofiber complex was evaluated using drug release kinetics via UV Spectrophotometry, Fourier-Transform Infrared Spectroscopy (FTIR), Scanning Electron Microscopy (SEM), and Differential Scanning Calorimetry (DSC). Drug loading, apparent solubility, and antioxidant activity were also evaluated. The study was successful in creating fiber-like encapsulation of the silymarin drug with strand diameters ranging from 5–7 μm, with results showing greater silymarin release in Simulated Intestinal Fluid (SIF) compared to Simulated Gastric Fluid (SGF). Moving forward, this study aims to provide future insight into the formulation of drug–polymer complexes for IBD treatment and targeted drug release using electrospray and microencapsulation.

## 1. Introduction

Inflammatory bowel disease (IBD) encompasses chronic or recurrent inflammation of the gastrointestinal (GI) tract, primarily including ulcerative colitis (UC), which affects the colon, and Crohn’s disease (CD), which can involve the entire GI tract [1]. IBD predominantly affects individuals aged 20–40 years, imposing a significant socioeconomic burden. The relapse rate for IBD is 50–80%, requiring long-term management and increasing healthcare costs. There have been numerous pharmacological medical therapies which have been proposed for IBD, like Salicylates, glucocorticoids and immunosuppressives [2], but its medical management remains challenging, and investigations into novel treatments such as herbal extracts containing polyphenols as a new strategy for the management of IBD are currently in the center of attention [3,4].

Pharmacological studies with experimental animals over the past decade indicate that polyphenols are effective in preventing and alleviating complications of ulcerative colitis (UC) [5]. Among the various botanicals examined, curcumin, the primary component of turmeric, is a highly researched phytochemical for treating inflammatory bowel disease (IBD) and other conditions, due to its non-toxic, antioxidant, anti-inflammatory, and cytoprotective properties. Numerous preclinical studies have shown that curcumin improves survival rates and reduces discomfort caused by chemical ulcerogens, scavenges free radicals, influences multiple signaling pathways, and inhibits enzymes like COX-1, COX-2, and LOX. Clinical studies suggest that curcumin can inhibit clinical relapse in IBD patients [6,7]. Similarly, resveratrol, found in grapes and other berries, offers multiple pharmacological benefits, including preventing colitis, reducing colorectal carcinogenesis, and improving disease prognosis by modulating immune cell numbers and inflammatory markers [7,8]. Quercetin, a flavonoid present in various fruits and vegetables, exhibits free-radical scavenging, antioxidant, and anti-inflammatory properties, effective in the early stages of TNBS-induced colitis [7,9]. Kaempferol, found in many edible plants, has antioxidant, anti-inflammatory, antimicrobial, anticancer, and other protective activities, significantly reducing DSS-induced colitis [7,10]. Ellagic acid from berries and pomegranates, rutoside from buckwheat, green tea polyphenols, and grape seed polyphenols have all been shown to reduce the severity of dextran sodium sulfate and 2,4,6-trinitrobenzene sulfonic acid-induced colitis, alleviate oxidative stress, and prevent inflammation-associated colon carcinogenesis, due to their anti-inflammatory and antioxidant properties [11,12]. Collectively, these phytochemicals exhibit significant potential in treating IBD and other inflammatory conditions.

Silymarin, a herbal extract from milk thistle seeds, with a complex composition of four flavonolignans, silybin, isosilybin, silydianin and silychristin, is most known for its hepatoprotective qualities [13]. The drug has also been shown to positively treat the colon, especially in cases involving ulcerative colitis [14]. Additionally, silymarin has antioxidant properties, enhances antioxidant enzyme activity, interacts with cell membranes to prevent lipid deformity, and manages toxic stress. Additionally, silymarin exhibits anti-inflammatory effects by stabilizing mast cells, inhibiting neutrophil infiltration, decreasing adhesion molecules, downregulating leukotriene and prostaglandin synthesis, and inhibiting key cytokines involved in inflammatory responses such as interleukin (IL)-6 and transforming growth factor (TGF)-β, which are key cytokines in the differentiation of regulatory T cells CD4 + CD25-Foxp3- and T helper (Th) 17 cells; it also has inhibitory effects on inducible nitric oxide (NO) synthase activity [15,16,17,18,19]. The therapeutic and biochemical effects of silymarin that have been studied show promising evidence of its positive influence on various types of cells. In cases of Amanita mushroom poisoning, silymarin has been shown to be an active treatment and to reduce patient mortality [5,20]. Additionally, silymarin has demonstrated a strong anti-angiogenesis effect on the colon cancer cell line in vitro [21]. Due to the significant antioxidant, anti-inflammatory, and immunomodulatory properties of silymarin, we investigated its potential protective antioxidant activity through an assay. However, one problem with silymarin is that the drug is poorly soluble in water [22], leading to investigations involving drug encapsulation for the targeted release of silymarin in the body. Studies have accomplished extremely small particle size (461 ± 173 nm) with a combination of polyvinylpyrrolidone and silymarin by homogenization via nanoprecipitation, demonstrating increased solubility and enhanced antioxidant activities.

Antioxidant activities were demonstrated by the 1,1-diphenyl-2-picrylhydrazyl radical (DPPH) method [10]. Biopolymer-based and lipid-based systems have also been investigated [23], as new procedures involving nano-encapsulation of silymarin can help the drug bypass factors, limiting its poor oral consumptive properties for more effective drug loading and drug release. While silymarin nanoparticles have also been prepared using emulsion solvent evaporation and freeze-drying methods to improve solubility [24], an electrospray method for improved silymarin solubility and pH-dependent targeted drug release in the colon has not been developed. The hepatoprotective activity of silymarin nanoformulations has been confirmed with a mixture of soy lecithin, with an average diameter between 138.9 nm and 1155 nm, indicating improved therapeutic efficacy of silymarin with a nano approach [25]. Silymarin nanoencapsulations even been shown to have antimicrobial applications when formulated in combination with chitosan (WCS) and poly-γ-glutamic acid (γ-PGA) for improved water solubility in food additives and food packaging, compared to unencapsulated silymarin [26]. The therapeutic and biochemical effects of silymarin that have been investigated show promising evidence of its positive influence on various types of cells. For example, silymarin has been shown to increase superoxide dismutase activity in humans, which plays a key role in defending cells from free radicals. While various delivery methods for silymarin such as biopolymer-based and lipid-based systems have been investigated [7], new procedures involving nano-encapsulation and micro-encapsulation of silymarin can help the drug bypass factors limiting its poor oral consumptive properties for more effective drug loading and drug release.

Electrospraying (electro-hydrodynamic spraying) is a process for generating droplets by applying an electric field. In this process, a drug/polymer solution is subjected to an electric field flowing out from a capillary nozzle maintained at high potential [27]. When the electric field attains a critical value, a jet is formed. The electric field then causes deformation and distribution of the jet into droplets. Indeed, electrospraying and electrospinning are based on the same principles, except that the jet formed in electrospinning does not break into droplets but produces a micro- or nanofiber [28]. Generally, both nanofibers and nanoparticles function to increase bioavailability, due to their unique physicochemical properties. The main difference in, and reason for, the use of nanofibers with silymarin is that they typically offer a larger surface area-to-volume ratio compared to nanoparticles; this can enhance bioavailability and provide controlled release, better stability, and improved interaction with biological tissues, which is necessary for pH-dependent target release in the colon surface area and a greater sustained-release profile of nanofibers. The major advantage of electrospun fibers is that the system can be set up for the release and delivery of multiple drugs, with multi-polymer and drug systems [29]. Conversely, nanoparticles provide a more uniform and potentially faster release profile due to their smaller size, which can be advantageous for rapid-onset applications. The electrospraying procedure has many advantages when compared to nanoprecipitation, such as producing smaller droplet sizes with narrow distribution, the absence of droplet agglomeration and coagulation, since charged droplets are self-dispersing in the space, and easy control of motion and deposition efficiency of charged droplets. Essentially, highly charged potential forces can result in dividing charged droplets into smaller droplets. This is defined as Coulomb fission of the droplets, which causes original dispersed droplets to form many smaller, more stable droplets. The bulk forces include electro-dynamic forces, inertia, gravity, and drag forces, which are the physics governing electrospraying. When the induced droplet flows and deforms, (as a Taylor cone-jet), surface stresses act against surface tension including electro-dynamic stress (proportional to the charge density on the surface of the jet, and on the local electric field), pressure differential across the jet–air interface, and stresses due to liquid dynamic viscosity and inertia [30]. The literature reviews on the process and its application in the pharmaceutical field include the following: nano/microparticles particles were produced for drug delivery such as PLGA nanoparticles with Paclitaxel [31], electrosprayed coenzyme Q10 in copovidone (Kollidon^®^ VA64) [32], streptokinase-loaded PLGA nanoparticles [33], encapsulating drugs such as paclitaxel and topotecan in PLGA-chitosan [34], resveratrol in Hyaluronic acid-ceramide and soluplus [35], Doxorubicin in PVA-silk fibroin [36], Oridonin in PLGA [37], encapsulating DNA, and enzymes such as PEG and trehalose were added to PLGA to prepare vascular endothelial growth factor (VEGF)- or bone morphogenetic protein 7 (BMP-7)-loaded carriers via electrospraying [38], polymeric coatings on medical implants [39], and biomolecule carriers for tissue and bone regeneration. These are applications explored for electrospraying techniques [40,41]. Polyvinylpyrrolidone and Sodium dodecyl sulfate loaded with silymarin-laden nanocontainers with a particle size of <1000 nm have been developed successfully for improved aqueous solubility y (26,432.76 ± 1749.00 μg/mL) and dissolution (~92% in 20 min), compared to plain drug powder [42].Cellulose acetate (CA) fibers with silymarin were made into fibers using the electrospinning method with 608 ± 133 nm diameter with 12.5 kV voltage application when the needle is 15 cm away from the spinning drum, and researched the stable drug release 120 min 1/1 phosphate buffer/methanol medium pH 7.4 at 37 °C [43]. In another study, polycaprolactone (PCL) loaded with silymarin with different concentrations (mainly 5, 7.5 and 10 wt%) were produced by electrospinning to develop a functional wound dressing. In vitro drug-release studies were conducted using phosphate-buffered solution (PBS) at pH 7.4 and 36 °C, and time-dependent release values were examined [44]. Eudragit^®^-based electrospray/spin optimization was conducted by employing Eudragit^®^ E PO and Chlorpheniramine Maleate, where 35% Eudragit E PO at a gap distance of 175 mm and a flow rate of 1 mL/h were identified as optimum conditions for fiber production [45] and for ketoprofen (KET)-loaded Eudragit^®^ L and Eudragit^®^ S nanofibers. The electrospinning technique for buccal administration to treat oral mucositis was used, where optimum conditions were a lower flow rate (0.5 mL/h), as higher flow rates led to thicker fibers and structural deformations, due to insufficient drying. A voltage of 15 kV and 20% *w*/*v* Eudragit^®^ S100 with 10% *w*/*v* KET demonstrated high drug loading efficiency [46]. In another study, Eudragit^®^ L 100 was electrospun with diclofenac sodium, where the Eudragit^®^ L 100 was set to 20% (*w*/*v*) in ethanol and DMAc in a 5:1 ratio, with an applied voltage of 10 kV, and a flow rate 1.0 mL/h. Different drug concentrations were tested: 9.1%, 16.7%, and 33.3% (*w*/*w*); higher drug concentrations were observed to increase the fiber diameter and potentially affect the surface morphology [47]. In another study, Eudragit^®^ S100 Nanofibers were prepared with Aspirin and a Eudragit^®^ S100 concentration of 15% *w*/*v*, with a polymer: drug ratio of 5:1, an applied voltage of 15 kV and an average flow rate of 1 mL/h, resulting in average diameter of 800 ± 110 nm [48]. In another study, Eudragit^®^ L100 fibers 25% (*w*/*v*) were mixed in N, N-Dimethylacetamide (DMAc), ethanol, and methanol, with an applied voltage of 12 kV and flow rate maintained at 2.0 mL/h [49]. Finally, another study employed 13 % *w*/*v* Eudragit^®^ S100 with 5-Fluorouracil, a mixture of ethanol and N,N-dimethylformamide (DMF) (8:2 *v*/*v*), along with other ingredients, with an applied voltage of 14.5 kV–16 kV and flow rate of 1.5 mL/h, with drug concentration maintained at 10% *w*/*v* [50].

In this study, the polymer Eudragit^®^ S100 was combined with silymarin to create a colon-targeted drug-delivery system using vertical electrospray technology. The resulting complex, termed the SILS100-Electrofiber complex, was formulated with an 11.2% ratio of silymarin to Eudragit^®^ S100. This complex represents the first successful micro-encapsulation of silymarin into fibers. The paper details the electrospray process and its optimization, characterizes the complex, and examines its pH-dependent release properties. The findings demonstrate that this formulation enhances aqueous solubility and improves the dissolution rate in the colon’s specific pH environment, indicating its potential for effective colon-targeted drug delivery.

## 2. Materials and Methods

### 2.1. Materials and Reagents

The experimental materials consisted of silymarin, an Antioxidant Colorimetric Assay Kit, and Spin-X^®^ Centrifuge Tubes purchased from Sigma Aldrich, St. Louis, MO, USA, Eudragit^®^ S100 donated from Evonik Industries, Essen, Germany, Spectrum Spectra/Por Float-A-Lyzer G2 Dialysis Devices (3.5–5 kD), Simulated Gastric Fluid (SGF) without pepsin and Simulated Intestinal Fluid (SIF) without pepsin purchased from Fisher Scientific Inc. located in Pittsburg, PA, USA.

### 2.2. Electrospray Setup

The SILS100-Electrofiber complex was fabricated using the E-Fiber EF050, Bollate (MI), Italy, an advanced electrospray setup provided by SKE Research Equipment, Bollate (MI), Italy. The process involved optimizing several parameters to ensure a stable fiber formation. These parameters included the type of organic solvent (methanol or acetone), applied voltage, needle gauge, the distance between the needle and the collector, the ratio of silymarin to Evonik Industries, Essen, Germany-Eudragit^®^ S100 solutions, and the flow rate of the solution. By fine-tuning these variables, a consistent and effective electrospun fiber was achieved. The electrosprayed fibers were collected in a glass jar and subsequently stored for detailed analysis.

### 2.3. Preparation of SILS100-Electrofiber Complex

To investigate the formation of Eudragit^®^ S100 fibers, solutions with varying weight percentages (wt%) of the polymer were prepared in acetone. These solutions were then electrosprayed onto aluminum foils. The resulting fibers were examined under a scanning electron microscope to analyze the morphology at each wt%. Additionally, the viscosity of the different wt% solutions was measured, as this property is crucial for effective fiber formation. For the development of the SILS100-Electrofiber complex, specific ratios of silymarin to Eudragit^®^ S100 were created, including 1:2, 1:5, 1:10, 1:15, and 1:20. These ratios were achieved by vortexing precise amounts of silymarin and Eudragit^®^ S100 in acetone. For instance, a 1:10 ratio was prepared by vortexing 0.0227 g of silymarin with 0.227 g of Eudragit^®^ S100 polymer in 2.2 mL of acetone for 15 min. Each of these solutions was then electrosprayed, and the resulting fibers were collected in a jar for further analysis. This thorough approach ensured the accurate preparation and characterization of the SILS100-Electrofiber complex.

### 2.4. Fourier-Transform Infrared (FTIR) Spectroscopy

The formation of the SILS100-Electrofiber complex was confirmed through Fourier-Transform Infrared (FTIR) analysis using the Agilent Cary 630 FTIR spectrometer (Agilent Technologies, Inc., Santa Clara, CA, USA) ATR methods. FTIR spectra were obtained for pure silymarin, Eudragit^®^ S100, a physical mixture of silymarin and Eudragit^®^ S100, and the SILS100-Electrofiber complex. These spectra were collected at a scanning speed of 4 cm^−1^ over a range of 4000 to 400 cm^−1^.

### 2.5. Scanning Electron Microscopy (SEM)

The detailed examination of the surface morphology and structural characteristics of the SILS100-Electrofibers was thoroughly analyzed using scanning electron microscopy (SEM) with an FEI Quanta 650 Scanning Electron Microscope from Bruker in Berlin, Germany. To enhance the imaging quality, the electrosprayed SILS100-Electrofibers were sputter-coated with a thin 10 nm layer of gold. This coating process was essential for improving the conductivity of the samples and obtaining clearer images. The fibers were carefully mounted on metal holders using conductive double-sided tape to ensure stability during the imaging process. SEM images were captured under various accelerating voltages, with a particular focus on 30.0 kV to achieve optimal resolution and detail.

### 2.6. Differential Scanning Calorimetric (DSC) Analysis

DSC analysis was performed to assess the solid-state characteristics of pure silymarin, Eudragit^®^ S100, a physical mixture of silymarin and Eudragit^®^ S100, and the SILS100-Electrofiber complex. This analysis was conducted using the DSC 3 STARe instrument from Mettler Toledo (Columbus, OH, USA). During the DSC analysis, thermograms were obtained at a scanning rate of 10 °C per minute, covering a temperature range from −20 °C to 300 °C. Liquid nitrogen was employed as a coolant to maintain precise temperature control throughout the analysis. By comparing the thermograms of the individual components, the physical mixture, and the SILS100-Electrofiber complex, the DSC analysis provided valuable insights into the thermal behavior, crystallinity, and potential interactions within the complex. This comprehensive thermal analysis was crucial for understanding the stability and compatibility of the components within the SILS100-Electrofiber complex.

### 2.7. Assessment of Drug–Polymer Complex Formation

The evaluation of the SILS100-Electrofiber complex involved dispersing both pure silymarin and the SILS100-Electrofiber complex in two different pH conditions: pH 1.4 (Simulated Gastric Fluid (SGF) without pepsin) and pH 7.4 (Simulated Intestinal Fluid (SIF) without pepsin) at 37 °C. During the experiment, samples were collected at 15 and 75 min. These samples were then centrifuged at 20,000 rpm for 20 min. After centrifugation, it was passed through Spin-X-UF concentrator tubes (Sigma Aldrich, St. Louis, MO, USA) with a 10 kDa membrane cutoff. The ratio (bottom/top) was calculated as the concentration of silymarin in the solution before (top of the tube) and after (bottom of the tube) passing through the 10 kDa cut-off membrane [14]. This ratio was determined by measuring the concentration of silymarin using a UV–visible spectrophotometer at a wavelength of 326 nm.

### 2.8. Silymarin Loading in the SILS100-Electrofiber Complex

The determination of silymarin loading, expressed as milligrams of silymarin per milligram of the total drug–polymer complex, was performed using a specific extraction and analysis procedure. First, 25 mg of the SILS100-Electrofiber complex was dissolved in 40 mL of 1X PBS buffer. This solution was placed in a Spectrum Spectra/Por Float-A-Lyzer G2 Dialysis Device (Sigma Aldrich, St. Louis, MO, USA) with a molecular weight cutoff of 3.5–5 kDa. Silymarin extraction was carried out over a period of 4 h. After this period, the amount of silymarin in the extract was quantified using a UV–visible spectrophotometer at a wavelength of 326 nm. This method provided an accurate measurement of the silymarin content within the complex, ensuring the proper evaluation of drug loading efficiency in the SILS100-Electrofiber complex.

### 2.9. Apparent Solubility of the SILS100-Electrofiber Complex

This investigation is adopted from Reference [51], where the term “apparent solubility” refers to the solubility measured after a 4 h incubation period, which does not account for thermodynamic or equilibrium solubility. To assess this parameter, solubility studies were conducted at four different pH levels: pH 1.2, 4.0, 6.0, and 7.4. For each pH condition, 25 mg of the SILS100-Electrofiber complex was dispersed into a corresponding vial containing the appropriate buffer solution. The samples were then incubated at 37.5 °C for 4 h with continuous stirring at 200 rpm to ensure thorough mixing and interaction between the complex and the buffer solution. This method allowed for the evaluation of the apparent solubility of the SILS100-Electrofiber complex across different pH environments, providing valuable insights into its potential behavior and effectiveness in various physiological conditions.

### 2.10. pH-Dependent Solubility of the SILS100-Electrofiber Complex

A precise determination of the release profile and solubility behavior of the SILS100-Electrofiber complex in various pH environments was evaluated using Spectrum Spectra/Por Float-A-Lyzer G2 Dialysis Devices with a molecular weight cutoff of 3.5–5 kDa, and various buffer solutions with pH values of 1.4, 4.0, 6.0, and 7.4. For each pH condition, 25 mg of the SILS100-Electrofiber complex was suspended in the appropriate buffer and maintained at 37 °C. To measure the concentration of silymarin released from the SILS100-Electrofiber complex, a UV–visible spectrophotometer was employed at a wavelength of 326 nm. Samples were periodically collected over different time frames, depending on the pH: 2 h for pH 1.4, 4 h for pH 4.0 and pH 6.0, and 10 h for pH 7.4. A standard curve prepared using silymarin dissolved in methanol and acetone gave the same results in terms of wavelength maximum. This was utilized to calculate the concentration of silymarin in the samples.

### 2.11. Antioxidant Assay of the SILS100-Electrofiber Complex

To investigate the antioxidant potential of the complex, highlighting its potential therapeutic benefits for the GI tract, an antioxidant activity assay of the SILS100-Electrofiber complex was conducted using the Antioxidant Colorimetric Assay Kit purchased from Sigma Aldrich located in St. Louis, Missouri, USA. For this assay, 25 mg of the SILS100-Electrofiber complex was placed into 10 mL of either simulated gastric fluid (pH 1.2) or simulated intestinal fluid (pH 7.4). The samples were then incubated for four hours at a temperature of 37.5 °C, with continuous stirring at 300 rpm to ensure thorough mixing and interaction.

## 3. Results and Discussion

Eudragit^®^ S100 is an anionic copolymer based on methacrylic acid and methyl methacrylate in a 1:1 ratio, with a dissolution threshold at pH of 7.0 [52]. Due to its acidic nature, ES100 has very low solubility in water and acids, but ionization of their carboxylic groups by the use of alkaline solutions could be a possible approach to enhance its aqueous solubility [53].

Figure 1a illustrates the schematic diagram for the electrospray process used to fabricate the SILS100-Electrofiber complex, as well as the characteristics of the solid fibers produced. Figure 2 provides close-up views of the fibrous material being manipulated with tweezers. Specifically, Figure 2A shows a cluster of white solids directly collected in the jar. Figure 2B presents a twisted cluster of fibers, while Figure 2C depicts the fibers further stretched using the tweezers.

The optimization of electrospray parameters for fabricating the drug–polymer complex involved careful adjustment of several key variables, which are discussed further. The chain-entanglement concentrations of Eudragit^®^ S100 E-EPO ranging from 0.5 to 35 wt% were electrospun and the solution viscosity was characterized to determine a chain-entanglement concentration as described by Kong and Ziegler [54]. This is defined as the intersection point of the two fitted lines that represent the untangled and entangled regions, as illustrated in Figure 3A. The chain-entanglement concentration, which is crucial for the formation of smooth fibers and the viability of the electrospinning process, was determined to be 10.2% *w*/*v* for Eudragit^®^ S100. To complement the viscosity measurements, SEM images of the electrospun Eudragit^®^ S100 fibers are presented in Figure 4A–G.

Figure 4A illustrates that at lower concentrations of 0.5–1%, the electrospraying process is dominant. As the polymer concentration increases, a transition from electrospraying to electrospinning occurs. Fiber formation begins at 5% Eudragit^®^ S100, as seen in Figure 4B, initially displaying a bead-on-string morphology. However, smooth fibers are not observed until the concentration reaches 10% (Figure 4C). Figure 4D–G show that as the polymer concentration continues to increase, the fibers become smoother and increase in diameter.

This increase in fiber diameter is attributed to the higher amount of polymer being dispensed through the needle, as shown in Figure 3B. Additionally, the fibers exhibit a ribbon-like appearance, likely due to the rapid evaporation of the solvent, causing the fibers to collapse and appear flat [55].

The drug/polymer ratio of the SILS100-Electrofiber complex was optimized depending on whether the ratios were able to be electrosprayed to obtain bead-free fibers. They were successfully produced by electrospinning at a voltage of 25 kV, with a flow rate set to 2.0 mL/h. The distance from the needle to the collector was precisely maintained at 5.0 cm, using an 18-gauge needle. The ideal voltage was determined at 25 kV, which ensured the charged particles were sprayed smoothly and consistently. Lower voltages did not produce a consistent spray, resulting in a conglomeration of material in the needle. The ideal flow rate was determined at 2.0 mL/h with an 18-gauge needle because of the continuous flow at such a high voltage. Flow rates higher than 2.0 mL/h caused liquid to spray out in addition to the fibrous strands, indicating there was not enough time for all the particles to become charged. Flow rates lower than 2.0 mL/h did not produce a consistent spray, and resulted in sputtering at the needle. The distance from the needle to the collector was kept at 5.0 cm to collect and contain the final SILS100-Electrofiber complex product in a glass container. With the chain-entanglement concentration set to more than 10.2%, the concentration of Eudragit^®^ S100 was set to 11.2 wt% for all the formulations. The ratio of drug to polymer was altered (1:2, 1:5, 1:10, 1:15 and 1:20) to increase silymarin loading and surface area without impairing silymarin solubility. The 1:10 ratio of silymarin to Eudragit^®^ S100 was selected due to its bead-free fibers, as, compared to the 1:2 and 1:5 ratios, it did not produce fibrous strands and maintained the fibers’ liquid form. Additionally, the 1:15 and 1:20 ratios did not produce fibrous strands and sprayed into the container as liquid particles that were not cohesive. Another factor was to optimize the solvents that the silymarin and Eudragit^®^ S100 polymer will dissolve. Solvents such as methanol did dissolve the drug but did not dissolve the polymer; hence, acetone was used based on its ability to dissolve both the drug and the polymer. This solvent also exhibits quick drying in the electrospray setup.

FTIR spectroscopy is a valuable tool for identifying interactions between different components in a formulation. The FTIR plot in Figure 5 shows the vibrational spectroscopic peaks for a 50/50 (1:1 ratio) physical mixture of silymarin and Eudragit S100, as well as for the SILS100-Electrofiber complex. A notable difference between the physical mixture and the electrofiber is the C-H stretch at 2900 cm^−1^; the broad peak around 3500 cm^−1^ in the electrofiber indicates the possibility of hydrogen bonds forming between the phenolic -OH group of silymarin and the backbone of Eudragit^®^ S100. Additionally, the electrofiber exhibits peaks representing the -C=O group from the Eudragit^®^ S100 backbone. The FTIR spectrum of pure silymarin shows characteristic absorption peaks at 3647 cm^−1^ (O-H), 2876 cm^−1^ (C-H), 1643 cm^−1^ (C=O), and 1513 cm^−1^ (aromatic C=C) [56], and pure Eudragit S100 polymers contain 1446.14 cm^−1^ (carboxylic acid) and 1728 cm^−1^ (ester groups) [57,58]. The FTIR plots confirm the interaction between silymarin and Eudragit S100 and the retention of their characteristic peaks throughout the production process.

SEM images shown in Figure 6 depict a fiber diameter of 5–7 μm and fibrous entanglement of the SILS100-Electrofiber complex. Light microscope images are additionally shown depicting the fibrous strands of the SILS100-Electrofiber complex.

DSC analysis is depicted in Figure 7a, with thermograms showing the DSC profile of a pure Eudragit^®^ S100 wide endothermic peak at 90 °C and 216.2 °C [59] which did not exhibit a distinct melting point peak, indicating the amorphous nature of the polymer. Figure 7b shows a broad endotherm at about 90 °C indicating the melting of crystalline silymarin and a glass transition temperature of around 150 °C [58,60]. Figure 7d shows the thermogram for the SILS100-Electrofiber complex with characteristics of both silymarin and the Eudragit^®^ S100 which are missing in the physical mixture of the two components (Figure 7c).

The existence of complexation between silymarin and the polymer in aqueous solution was confirmed by passing the aqueous solution of the SILS100-Electrofiber complex through a known molecular weight cut-off (10 kDa). As shown in Figure 8a, which illustrates the bottom/top ratio, this is defined as the concentration of soluble silymarin that passed through the 10 kDa cut-off membrane (bottom) relative to the concentration before filtration (top). This ratio serves as an indicator of the extent of silymarin release from the SILS100-Electrofiber complex [51]. Measurements were taken at two pH levels, 1.4 and 7.4, at both 15 min and 75 min. The data reveal a substantial increase in the bottom/top ratio at pH 7.4 after 75 min, indicating a significantly higher release of silymarin in this environment. Figure 8b focuses on the silymarin concentration released at pH 1.4, comparing the SILS100-Electrofiber complex to pure silymarin (control). The results at both 15 min and 75 min show minimal release of silymarin from both the complex and the control. This suggests limited solubility and release of silymarin at the acidic pH typical of the gastric environment. Figure 8c presents the silymarin concentration released at pH 7.4, again comparing the SILS100-Electrofiber complex to pure silymarin (control). Notably, there is a significantly higher release of silymarin from the SILS100-Electrofiber complex at 75 min compared to the control. This finding indicates enhanced solubility and release of silymarin in the alkaline pH typical of the intestinal environment. Figure 8d visually depicts the filtration setup using SpinX^®^ tubes for both the SILS100-Electrofiber complex and pure silymarin. The diagram highlights the fact that the complexation with Eudragit^®^ S100, which has an approximate molecular weight of 125 kDa, prevents the silymarin from passing through the 10 kDa cut-off membrane. In contrast, soluble silymarin does filter through, enabling the quantification of the released drug.

The Spin-X assessment results suggest that the SILS100-Electrofibers do not release silymarin at lower, gastrointestinal pHs, and are more likely to release silymarin at higher pHs. A standard curve prepared using silymarin in methanol (or acetone) was used to calculate the concentration of silymarin at 326 nm with an R^2^ value of 0.9919 and the line equation of y = 1.5606x − 0.0294. Pure silymarin was employed as a control and run in parallel with the SILS100-Electrofiber complex. Silymarin loading in the SILS100-Electrofiber complex was calculated as 34% ± 0.6. The concentration of silymarin extracted was 0.35138 mg/mL at pH 7.4 after 4 h.

The apparent solubility of the SILS100-Electrofiber complex is demonstrated in Figure 9a at four different pH levels: 1.2, 4.0, 6.0, and 7.4. In pH 1.2 and 4.0, the SILS100-Electrofibers did not dissolve but were rather pulled apart, causing the white film on top of the solution, looking like foam in the bottle, and only at the higher physiological pH 7.4 are the SILS100-Electrofibers soluble as shown as a cloudy solution. Figure 9b shows the pure silymarin drug at various pHs, and it is seen that the drug remains at the bottom of the vials. This shows that the apparent (kinetic) solubility of the SILS100-Electrofiber complex, which was estimated to be pH 7.4 for 4 h, is significantly enhanced.

Figure 10 shows the release of the SILS100-Electrofiber complex in vitro characteristics across different pH levels over time. The graphs show the percentage of drug release from the beads in various buffer solutions, specifically pH 1.4, 4.0, 6.0, and 7.4, over a period extending to 1440 min (24 h). Initially, the release of the drug is minimal at lower pH levels (1.4 in SGF with pepsin and 4.0), indicating that the SILS100-Electrofiber complex effectively retains the drug in highly acidic conditions. This is crucial for medications intended to be released in less-acidic environments like the colon. At pH 1.4, the drug release remains low, demonstrating that the coating with Eudragit^®^ S-100 successfully prevents premature drug release in the stomach’s acidic environment. The release remains controlled and minimal until the pH reaches 6.0, showing a slight increase in drug release. The percentage release starts to increase significantly as the pH rises, particularly after 480 min, when the pH is 7.4. A substantial increase in drug release is observed at pH 7.4, where most of the drug is released, indicating that the SILS100-Electrofiber complex is designed to release the drug in the colon, where the pH is typically higher. The cumulative release reaches nearly 100% at the end of the 1440 min, showing that the drug delivery system is effective in achieving targeted release in the desired pH environment.

The procedure reflects the sequential pH conditions encountered during the passage of food or drugs through the digestive system, including the stomach (pH 1.4), small intestine (pH 4–6), and large intestine/colon (pH 7.4). Figure 7 also conveys this release by percentage, reflecting that after 2 h at pH 1.4, 6% of the SILS100-Electrofiber complex was released. Then, after 4 h at pH 4.0, a total of 10% had been released, and after 4 h at pH 6.0, a total of 42% had been released. The remaining 58% was released at pH 7.4 over the following 16 h. Additionally, the release at pH 7.4 increased over time, while the other pH values generally exhibited stagnant drug release over time, in comparison. This indicates that even more of the drug may be released at pH 7.4 after 16 h.

The antioxidant activity of the SILS100-Electrofiber complex is depicted in Figure 11. The SILS100-Electrofiber complex demonstrated twice the antioxidant activity at pH 7.4 compared to pH 1.4. These findings suggest that the SILS100-Electrofiber complex has pH-dependent antioxidant activity, with enhanced activity in intestinal conditions compared to gastric conditions. This could have implications for the efficacy of the drug in scavenging free radicals and protecting against oxidative stress in different regions of the gastrointestinal tract.

## 4. Discussion

Ultimately, the main findings from this study demonstrate that the SILS100-Electrofiber complex exhibits a slow-release profile in acidic conditions (similar to the stomach) and a significantly higher release profile in pH conditions resembling those of the large intestine and colon. This highlights the targeted-delivery capabilities of the SILS100-Electrofiber complex in the large intestine and colon using the microencapsulation of silymarin with Eudragit^®^ S100 via the electrospray procedure. The targeted-release profile of the SILS100-Electrofiber complex is a significant advancement in drug-delivery systems, particularly for treatments requiring localized release in the lower gastrointestinal tract. Therefore, the SILS100-Electrofiber complex is intended for administration via a capsule, as the drug can bypass the acidic conditions of the stomach if it is consumed orally. Additionally, this study underscores the importance of considering pH conditions in the design and optimization of drug formulations for targeted release and efficacy, thus opening new possibilities for improving the therapeutic efficacy of drugs like silymarin, which are sensitive to the acidic conditions of the stomach or require localized action in the large intestine and colon. This approach could be potentially beneficial for treating various gastrointestinal ailments, ensuring that the active compound is delivered effectively and efficiently to the intended site of action, thereby enhancing overall treatment outcomes.

## Figures and Tables

**Figure 1 bioengineering-11-00864-f001:**
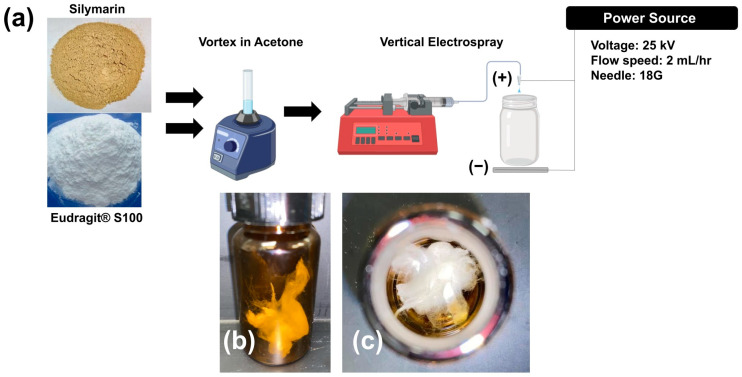
(**a**) Schematic image of the experimental procedure for the synthesis of the SILS100-Electrofiber complex (1:10 silymarin: Eudragit^®^ S100 ratio) and the electrospray set-up. (**b**) Side view of SILS100-Electrofiber complex after electrospray procedure (1:10 silymarin: Eudragit^®^ S100 ratio) and (**c**) top view of SILS100-Electrofiber complex after electrospray procedure (1:10 silymarin: Eudragit^®^ S100 ratio), showing the solid nature of the final product.

**Figure 2 bioengineering-11-00864-f002:**
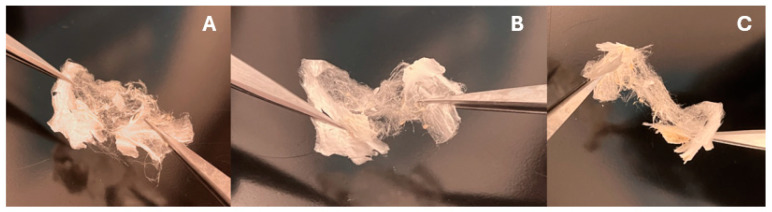
Images showing the fibrous textures of the SILS100-Electrofiber complex after the electrospray procedure (1:10 silymarin: Eudragit^®^ S100 ratio), showing the solid nature of the final product. (**A**) Complex after the electrospray. (**B**) the complex is twisted using a pair of tweezers. (**C**) Stretching the complex.

**Figure 3 bioengineering-11-00864-f003:**
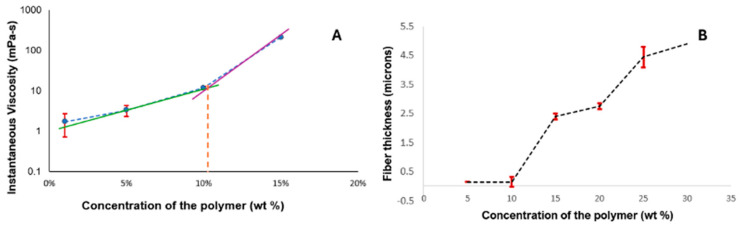
Minimum fiber-concentration measurement. (**A**) Viscosity measurement of the different concentrations. (**B**) Plot of concentration of the polymer and the increasing fiber thickness, showing that there were no fibers formed at 1 wt% concentration of the polymer, and that the minimum concentration required to form stable fibers was 10 wt%.

**Figure 4 bioengineering-11-00864-f004:**
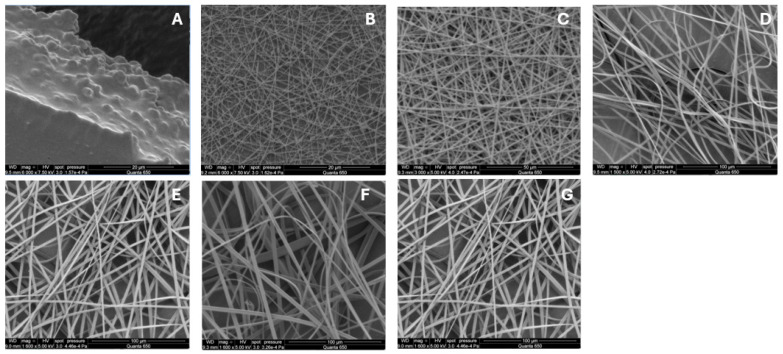
Scanning Electron Microscope (SEM) images of electrospun Eudragit^®^ S100 fibers: (**A**) 1, (**B**) 5, (**C**) 10, (**D**) 15, (**E**) 20, (**F**) 25, and (**G**) 30% w. All fibers were processed at applied voltages of 25 kV, with a flow rate set to 2.0 mL/h, and distance from the needle to the collector was precisely maintained at 5.0 cm, using an 18-gauge needle.

**Figure 5 bioengineering-11-00864-f005:**
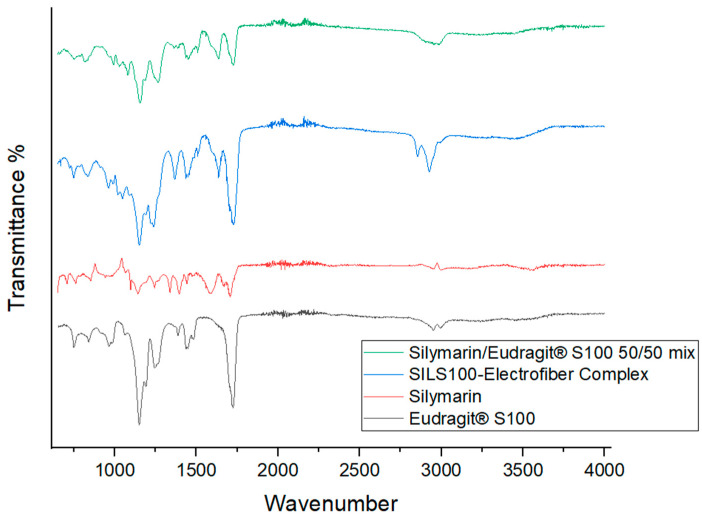
Fourier-transform infrared (FTIR) analysis of the silymarin/Eudragit^®^ S100 50/50 mix, SILS100-Electrofiber complex, silymarin, and Eudragit^®^ S100.

**Figure 6 bioengineering-11-00864-f006:**
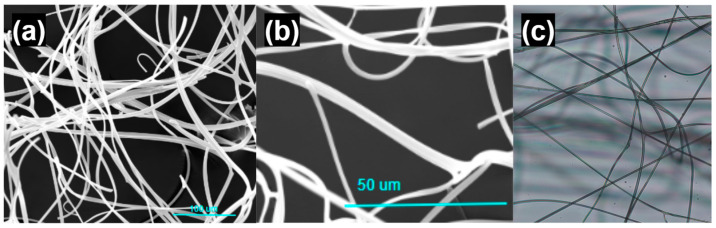
Microscope images of the SILS100-Electrofiber complex. (**a**) Scanning electron microscope image, scale bar of 100 microns. (**b**) Scanning electron microscope image, scale bar of 50 microns. (**c**) Light microscope image.

**Figure 7 bioengineering-11-00864-f007:**
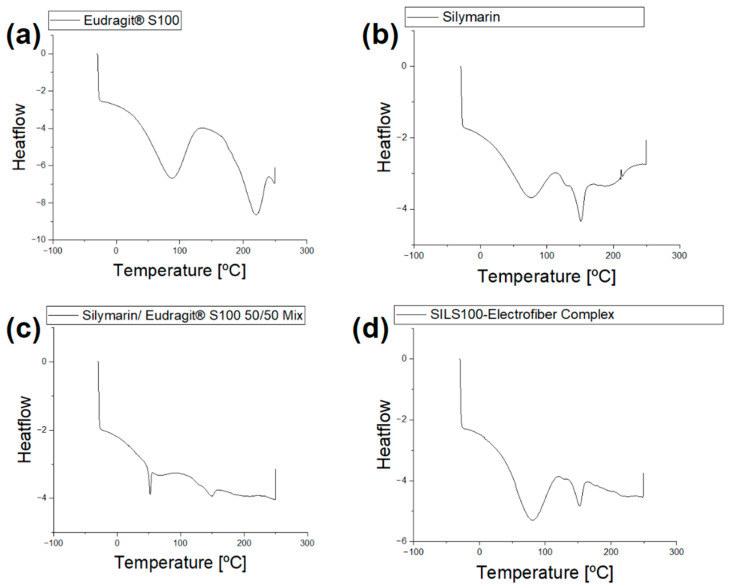
(**a**) DSC analysis of Eudragit^®^ S100. (**b**) DSC analysis of silymarin. (**c**) DSC analysis of the physical mixture of Eudragit^®^ S100 and silymarin (50/50 mix). (**d**) The DSC analysis of the physical mixture of the SILS100-Electrofiber complex.

**Figure 8 bioengineering-11-00864-f008:**
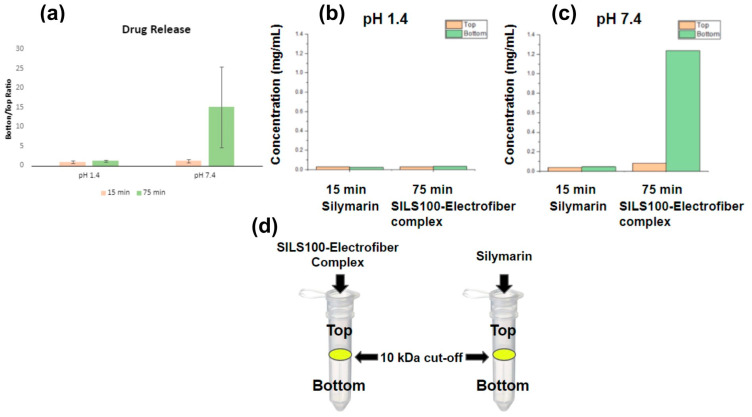
Assessment of SILS100-Electrofiber complex using SpinX^®^ Tubes. The bottom/top ratio is defined as the concentration of soluble silymarin that passed through the 10 kDa cut-off membrane (bottom) to the concentration before filtration (top). The higher ratio indicates a greater release of silymarin from the SILS100-Electrofiber complex. (**a**) Combined data from pH 1.4 and pH 7.4. (**b**) The ratio comparison of the SILS100-Electrofiber complex to the silymarin (control) at pH 1.4. (**c**) The ratio comparison of the SILS100-Electrofiber complex to the silymarin (control) at pH 7.4. (**d**) Visual representation of the 10 kDa cut-off SpinX^®^ tubes with the SILS100-Electrofiber complex and silymarin. Complexation with the Eudragit^®^ S100 with ~125 kDa size prevented silymarin filtering through the 10 kDa cutoff membrane.

**Figure 9 bioengineering-11-00864-f009:**
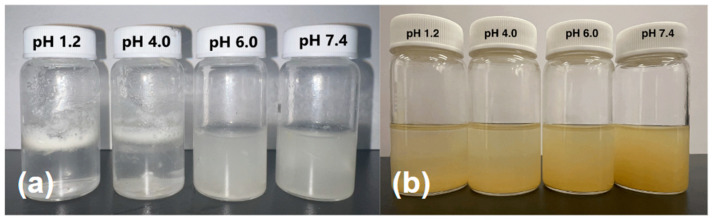
(**a**) Apparent solubility of the SILS100-Electrofiber complex, showing low solubility at pH 1.4 and higher solubility at pH 7.4. (**b**) Apparent solubility of the silymarin drug at different pHs, showing moderate solubility across all pH values tested.

**Figure 10 bioengineering-11-00864-f010:**
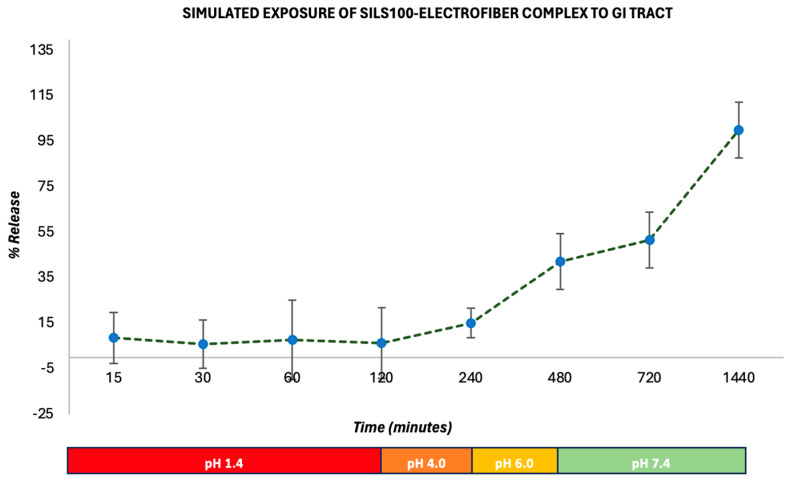
In vitro drug-release kinetics for the SILS100-Electrofiber complex at different pHs, simulating the GI tract.

**Figure 11 bioengineering-11-00864-f011:**
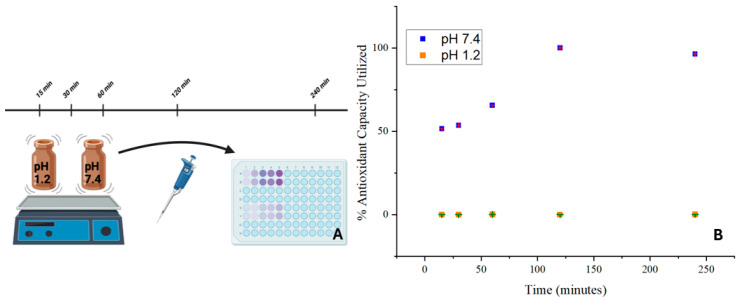
(**A**) Antioxidant assay of the SILS100-Electrofiber complex. (**B**) Antioxidant capacity of SILS100-Electrofiber complex.

## Data Availability

The raw data supporting the conclusions of this article will be made available by the authors on request.

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
