# Peer review of "Optimizing Production, Characterization, and In Vitro Behavior of Silymarin–Eudragit Electrosprayed Fiber for Anti-Inflammatory Effects: A Chemical Study"

_bioengineering, 2024, doi:10.3390/bioengineering11090864_

Round 1

Reviewer 1 Report (Previous Reviewer 1)

Comments and Suggestions for Authors

Authors have adressed most the comments However, it is suggested that although authors added examples of previous studies in introduction as suggested. But authors can write it presciely to reduce the length of introduction.  Secondly, calibration curve should also be prepared in same solvent in which release studies are carried for accuracy of results. If both are different results are not releiable.   

Comments on the Quality of English Language

Minor editing

Author Response

Authors have adressed most the comments However, it is suggested that although authors added examples of previous studies in introduction as suggested. But authors can write it presciely to reduce the length of introduction.  Secondly, calibration curve should also be prepared in same solvent in which release studies are carried for accuracy of results. If both are different results are not releiable.  

Thank you for bringing up these suggestions. The introduction has been decreased in length, and we believe the calibration curve is still appropriate given that many studies have been published with different solvents used for the calibration curve and analysis. Here are two examples:

Sun, Y.; Wang, J.; Gu, S.; Liu, Z.; Zhang, Y.; Zhang, X. Simultaneous Determination of Flavonoids in Different Parts of Citrus reticulata ‘Chachi’ Fruit by High Performance Liquid Chromatography—Photodiode Array Detection. Molecules 201015, 5378-5388. https://doi.org/10.3390/molecules15085378

March JG, Moukhchan F, Cerdà V. Application of in-vial membrane assisted solvent extraction to the determination of polycyclic aromatic hydrocarbons in seawater by gas chromatography–mass spectrometry. Analytica chimica acta. 2011 Jan 31;685(2):132-7.

Reviewer 2 Report (Previous Reviewer 2)

Comments and Suggestions for Authors

The manuscript has much improved and the authors have addressed all the comments, the only minor comments is to revise again the English language as there are some typo errors.

Also, the FTIR figure, one of the arrows pointing the C-H stretching vibration peak is on silymarin spectrum which is wrong, please revise. 

Author Response

The manuscript has much improved and the authors have addressed all the comments, the only minor comments is to revise again the English language as there are some typo errors.

Also, the FTIR figure, one of the arrows pointing the C-H stretching vibration peak is on silymarin spectrum which is wrong, please revise. 

Thank you very much for bringing up these suggestions. We have corrected the typo errors and revised the FTIR graph. 

This manuscript is a resubmission of an earlier submission. The following is a list of the peer review reports and author responses from that submission.

Round 1

Reviewer 1 Report

Comments and Suggestions for Authors

In this paper, authors have developed Silymarin S100-Electrofiber complex to address the issues faced silymarin administration. Paper need extension revision to overcome issues.

·       Title need revision, it is suggested to remove “Advancing Pharmaceutical Formulation”

·       Silymarin S100-Electrofiber complex is intended for oral administration. How it will be administered to the patients? Patch, powder, capsule etc.?

·       In introduction, previously reported Silymarin loaded nanoparticles are missing as few of examples are given below:  1- Silymarin nanoparticles through emulsion solvent evaporation method for oral delivery with high antioxidant activities, bioavailability, and absorption in the liver. 2 Characteristics and antioxidant activities of silymarin nanoparticles 3 https://doi.org/10.1016/j.jddst.2023.104378.   Many others are reported in literature.

·       How nanofibers loaded silymarin behaves in comparison to nanoparticles loaded silymarin? Owing to differences in morphology.

·       In material and method section, add a formulation chart for clarity. Designate a formulation code to each ratio and use them in subsequent testing for clarity.

·       Standard calibration curve should be prepared in same solvent where drug release was performed. Why only in methanol?

·       Please mention, which ratio is optimum formulation?

·       Error bars are missing in figure 2, 7and 9.

·       In figure 3, mention which ratio of fibers was used for FTIR studies? Is their any difference in results obtained for 50:50 mix of drug polymer and fiber complex for FTIR and DSC?

·       Results are not discussed properly. Improve the discussion part. No cross citation, add latest cross references.

Comments on the Quality of English Language

Minor

Reviewer 2 Report

Comments and Suggestions for Authors

The study describes the fabrication of silymarin -Eudragit microfibers (SILS100-

Electrofiber complex ) for targeted drug delivery of silymarin to the large intestine

for treatment of inflammatory bowl disorders. The release profiles in different pH

weas determined and an antioxidant assay was also performed. Two conditions in

the fabrication were optimized which are the type of solvent and ratio of silymarin to

polymer. The study is current, however several points have to be answered be the

authors before considering for publication.

1- The introduction lacks info about silymarin reported nanodrug delivery vehicles. Also

info about other polyphenolics reported nanodrug delivery vehicles. Also, previous

reports about electrosprayed drug delivery vehicles.

2- In materials and methods section subtitles should be added for each experiment.

3- In the experimental design, only the type of solvent and ratio of silymarin to Eudragit

were optimized but the parameters of the electrospraying (voltage, flow rate,

distance to collector) themselves were not varied, although they are also crucial in

determining the morphology of the final fibers.

4- The ratio bottom/top for calculating ratio of silymarin before and after passing

through spin concentrator tubes, is there a rational behind using this method and is

there a reference that could be added to this method in measurement? Please

explain the operation of the spin x tubes in more details.

5- The standard curve of silymarin was prepared in methanol, however the release

experiments were done in simulated gastric fluid and intestinal fluid which are

aqueous solutions, its better and more reliable if the standard curve is measured at

same conditions as the experiment.

6- In the determination of silymarin loading, how was silymarin extracted before

measurement as this is not mentioned.

7- It is mentioned in results that pure silymarin is released 0.9 times at pH 7.4 when

compared to pH1.2, the question is pure silymarin is released from what? Or the

authors just meant its dissolution? If this is the case pure silymarin is not

encapsulated , how come its dissolution is much less than when encapsulated within

a polymer??

8- Figure 2 a b and c lack error bars ?? how many times was the experiment done???

Where is the standard deviation?

9- The use of Spin x tubes and measurement of silymarin in bottom and top

compartments is not understood, is it a concentration gradient that derives the

diffusion of silymarin from top to bottom or what, please explain this experiment in

full details as it is not understood.

10- In FTIR results, its not clear which peaks are matching between silymarin drug and

the electrosprayed fibers, as also the peaks are matching with the eudragit polymer

itself. Please show clear differences. Also, please identify the major peaks in each

spectrum is due to which type of vibration? And highlight the peaks in the figure to

make it more clear.

11- Figure 6, the comparison of apparent solubility. The question is in the pure drug

experiment, is it done with the same amount of silymarin encapsulated in the fibers

or different amount???? As it should be the same amount for a fair comparison.

12- Figure 7 and 9 are without error bars.

13- Please indicate the basis behind the antioxidant assay and explain all its details in

the experimental section.

14- I have never seen a study with only 7 referenes, please add more up to date

references in the introduction, experimental and results section.

15- this  formulation is intended to be used as oral dosage form, but how will it be given?? directly as is or will incorporate in a capsule?? please indicate how will it be given orally??

Comments on the Quality of English Language

just minor editing of language

Reviewer 3 Report

Comments and Suggestions for Authors

The manuscript submitted by Liam Suskavcevic and co-authors deals with the nanoencapsulation of silymarin with Eudragit® S100 using an electrospray procedure (SILS100-Electrofiber complex) to improve dissolution for silymarin. The SILS100-Electrofiber complex was evaluated using various techniques, including silymarin release kinetics and antioxidant activity. Efficient delivery of poorly soluble drugs through oral systems remains a challenge. Therefore, in my opinion, the research topic covered in this work is important both from a practical and research point of view because it provides information on a simple synthesis method of the delivery system of a poorly soluble drug, silymarin, and the properties of the system. However, I do not think that the paper in this form exhibits the minimum recommended level for publication in Bioengineering. I recommend rejecting this manuscript in this form.

Several major objections affect the final conclusion, but I will mention only a few.

#1. 1. Introduction

The work quotes only 7 references. This number is insufficient considering the vast amount of research that has been conducted on silymarin, its chemical composition, separation method, activity, electrospraying, and its application in synthesising drug delivery systems and Eudragit® S100. This underscores the significance of your research. In my opinion, the Introduction and especially the literature review need to be improved. An in-depth discussion should be conducted regarding silymarin and its absorption from oral drug delivery systems.

#2. 2. Materials and Methods

Line 89-90:

It is not clear from the experimental part whether SGF and SIF fluids were used with or without pepsin.

Line 96:

The sentence “The rationale for selecting acetone as an organic solvent is based on its ability to dissolve both the drug and the polymer…” should include appropriate quotations regarding the solubility of silymarin and Eudragit® S100 in acetone.

Line 98:

What was the purpose of preparing a standard curve in methanol, especially since, in line 108, it states, "Silymarin polymer and Eudragit® S100 did not dissolve completely in methanol." Besides, what is synthetic silymarin? How was it obtained or where was it purchased?

There is no clear information on how to synthesize the final SILS100-electrofiber complex (the one which was further investigated). I do not understand what organic solvent was ultimately used and what the drug to polymer ratio was.

Line 112:

The description of the experiment concerning the dissolution/release of silymarin in SGF and SIF  is unclear. There is no information on the weight of the silymarin and SILS100-Electrofiber complex introduced into the SGF and SIF solutions or on the volume of the SGF and SIF solutions used in dissolution tests. Why was the Spin-X-UF concentrator tube used after centrifuging the solution? Why hasn't a silymarin standard curve been prepared for SGF and SIF? These issues need to be clarified.

Line 124:

Please add information on whether the presented FTIR spectra were taken using the ATR or transmission methods.

Line 147:

The description of the experiment regarding the solubility of silymarin in solutions of different pH is unclear. Firstly, there is no information on the volume of solutions used to dissolve silymarin from the SILS100-Electrofiber complex. Secondly, the composition of solutions with different pH is unknown. Were they buffers? Why did samples taken periodically over 2 hours require neither centrifugation nor application of the Spin-X-UF concentrator to assess silymarin concentration? These issues need to be clarified.

#3. 3. Results

The organization of the section entitled "Results" is chaotic, making the paper difficult to read. First, white material is visible in Figures 1b and 1c, but it is difficult to say that it is in the form of fibres. Therefore, the SEM images with appropriate discussion should be given after Figure 1. Next, the section concerning the characterisation of the SILS100-Electrofiber complex should be given. At the end of the "Results" section, the results of silymarin release should be presented.

Line 174:

Information on the appearance and texture of fibres (such as solid, cohesive, and flexible) cannot be derived from Figure 1.

Line 189:

Indeed, silymarin is better released from the SILS100-Electrofiber complex in SIF than in SGF, but I do not understand how was determined that the overall release of silymarin by the SILS100-Electrofiber complex in SGF (pH 1.4) was 4.7 times lower in concentration than the control using pure silymarin? These issues need to be clarified.

Figure 2a

Figure 2a is completely incomprehensible, but this is due to the lack of explanation of the use of the Spin-X-UF concentrator.

Figure 3:

It is accepted that FTIR spectra are presented with an inverted x scale (from 4000 - to 400 cm-1). In my opinion, this makes data analysis much easier for experts.

The bands discussed in the paper should be indicated in the figure.

The discussion of FTIR results needs to be improved. For example, there is no relevant discussion concerning the presence (or not) of hydrogen bonds between silymarin and Eudragit® S100.

Please explain the reason for measuring the FTIR spectrum for a physical mixture of silymarin and Eudragit® S100 50:50? In my opinion for a physical mixture of silymarin and Eudragit® S100, the ratio should be similar to in the SILS100-Electrofiber complex.

Figure 5:

I have the greatest objections about the DSC method and the discusson of the results. In fact, there is no discussion of the DSC results. Besides thermograms do not give information about

the efficacy of the SILS100-Electrofiber complex.  In the discussion, it should be mentioned already published works on the thermal degradation of silymarin and Eudragit® S100.

Similarly, there is no appropriate discussion about the results of the Antioxidant Capacity of the SILS100-Electrofiber complex.
